# Diabetic polyneuropathy with/out neuropathic pain in Mali: A cross-sectional study in two reference diabetes treatment centers in Bamako (Mali), Western Africa

Youssoufa Maiga [1,2,3]*, Salimata Diallo[2], Fatoumata dite Nènè Konipo[1], Oumar Sangho[1], Modibo Sangaré[1], Seybou H. Diallo[1,2], Saliou Mahamadou[4], Yann Péréon[5], Bernard Giumelli[6], Awa Coulibaly[2], Mariam Daou[2], Zoumana Traoré[7], Djeneba Sow Sylla[1,8], Mohamed Albakaye[2], Cheick Oumar Guinto[1], Madani Ouologem[6], Adama S. Sissoko[1], Hamar A. Traoré[1], Souleymane Papa Coulibaly[1], Philippe Damier[9], Nadine Attal[10], Julien Nizard[3,9,11]

1 Faculty of Medicine, University of Technical Sciences and Technologies, Bamako, Mali, 2 Department of Neurology, Gabriel Touré Teaching Hospital, Bamako, Mali, 3 Laboratory of Therapeutics (EA3826), Faculty of Medicine of Nantes, University of Nantes, Nantes, France, 4 Department of Diabetology, Gabriel Touré UHC, Bamako, Mali, 5 Reference Center of Neuromuscular Diseases Atlantique-Occitanie-Caraïbes, Hôtel-Dieu, UHC of Nantes, Nantes, France, 6 Faculty of Dental Surgery, University of Nantes, Nantes, France, 7 Department of Neurology, Hospital du Mali, Bamako, Mali, 8 Center for Combating Diabetes in Mali (CCD), Bamako, Mali, 9 Faculty of Medicine, University of Nantes, Nantes, France, 10 INSERM U 98, CET, CHU Ambroise Paré, 92100 Boulogne-Billancourt, France, 11 Federal Pain Palliative Care and Support, Laboratory of Therapeutics, Nantes UHC, Nantes, France

* youssoufamaiga@hotmail.com

**Data Availability Statement:** All relevant data are within the paper and its Supporting Information files.

## Abstract

### Introduction

Diabetic polyneuropathy (DPN) with or without neuropathic pain is a frequent complication of diabetes. This work aimed to determine the prevalence of diabetic polyneuropathy, to describe its epidemiological aspects, and to analyze the therapeutic itinerary of patients with DPN.

### Methods

This was a cross-sectional, descriptive study performed synchronously over six months at two major follow-up sites for patients with diabetes in Mali. DPN was diagnosed based on the Michigan Neuropathy Screening Instrument (MNSI). The neuropathic nature of the pain and the quality of life of patients were evaluated by the DN4 and the ED-5D scale, respectively. We used three (3) different questionnaires to collect data from patients (one at inclusion and another during the follow-up consultation) and from the caregivers of patients with DPN.

### Results

We included 252 patients with diabetes, and DPN was found to have a healthcare facility-based prevalence of 69.8% (176/252). The sex ratio was approximately three females for

**Funding:** The authors received no specific funding for this work.

**Competing interests:** The authors have declared that no competing interests exist.

every male patient. The patients were mostly 31 to 60 years of age, 83% had type 2 diabetes, and 86.9% had neuropathic pain Approximately half of the patients (48.3%) had autonomic neuropathy and they reported moderate to intense pain, which was mainly described as a burning sensation. The patients exhibited impaired exteroceptive and proprioceptive sensations in 51.7% of cases. The patients smoked tobacco in 3.4% of cases, while 36.6% of the patients were obese and had dyslipidemia. The caregivers clearly indicated that appropriate medications were not readily accessible or available for their patients with DPN.

## Conclusion

The healthcare facility-based prevalence of DPN with or without neuropathic pain was high in our cohort. These inexpensive and easy-to-use tools (MNSI, DN4) can be used to adequately diagnose DPN in the African context. In Mali, screening and early treatment of patients at risk of DPN should allow for a reduction of the burden of the disease, while caregivers need to be adequately trained to manage DPN.

## Introduction

The demographic changes taking place in developing countries are characterized by an increasing burden from non-transmissible diseases such as diabetes [1, 2]. The prevalence of diabetes in adulthood is 6% worldwide, with four out of five patients (80%) from low- or middle-income countries [3, 4]. Diabetes specifically impairs specific organs including the eyes, nerves, heart, and vasculature. Diabetes has conventionally been designated as being either type 1 or type 2 [5].

Compared to the Western world where "diseases of affluence" are rampant, Africa has long been thought to be significantly less affected. Interestingly, 7.1 million Africans are currently estimated to have diabetes, and this number could reach 18.6 million by 2030 [6]. Numerous socio-economic and cultural challenges will have to be overcome if the fight against diabetes in Africa is to be successful [7, 8].

A recent study in Mali has shown that the direct and indirect costs related to diabetes are not only higher than those of other chronic non-communicable diseases, but these costs are also much higher in relative terms than those in developed countries [9].

Diabetic polyneuropathy (DPN) with or without neuropathic pain is one of the most dreaded complications of diabetes [10]. DPN is a frequent and disabling complication, potentially affecting half of all patients with diabetes [11, 12]. DPN is defined as "a distal symmetric sensorimotor polyneuropathy, attributable to metabolic and microvascular changes resulting from chronic exposure to hyperglycemia" [13], or "the presence of symptoms or signs of peripheral nerve dysfunction in people afflicted with diabetes after exclusion of other causes" [14].

DPN has a considerable economic impact on the affected individuals and society in general. In the United States, the financial burden of the care allocated to DPN is estimated to be 10.9 billion dollars per year. Approximately 27% of the direct medical costs related to diabetes stem from neuropathic diabetes [14]. Surprisingly, DPN is nonetheless far from being considered a health priority in many countries [13, 15]. DPN is a serious complication of diabetes. Indeed, it can lead to foot ulceration and it inexorably progresses all the way to an increased risk of foot amputation with consequently a high mortality rate [16–18]. It is noteworthy that, compared to the other complications of diabetes, DPN is more often associated with depression

and anxiety disorders. It, therefore, requires early and effective treatment [19]. In practice, it is essential to develop effective intervention strategies to limit the physical and psychological impacts of DPN on patients [20, 21]. Most experts predict that DPN could well have severe consequences for health systems worldwide in general and in Africa in particular [22].

Epidemiological and clinical aspects as well as the quality of life of the patients with DPN have been understudied in Africa, especially French-speaking Western African countries. This study aimed to fill this knowledge gap in two ways by: (i) describing epidemiological aspects of diabetic polyneuropathy and (ii) analyzing the therapeutic itinerary of patients with DPN in Mali.

## Materials and methods

### Operational definitions

**Polyneuropathy.** Polyneuropathy was defined as the presence of symptoms and signs compatible with distal symmetric peripheral neuropathy that had been progressing for more than three (3) months [23].

**Proprioception impairment (PI).** This was considered to be present in patients with one or more of the following signs: hypopallesthesia to a tuning fork, difficulty localizing the big toe (arthrokinetic reflex), and a positive Romberg's sign.

**Objective exteroception impairment (OEI).** This was considered to be present in patients with one or more of the following signs: numbness, hyperesthesia to touch, and hypoesthesia or insensitivity to heat (a tube of hot water) or to cold (an ice pack).

**Autonomic neuropathy (AN).** The screening for autonomic neuropathy (AN) was purely clinical based on an interview, physical examination, and available non-invasive clinical tests, as defined by the American Diabetes Association [13]. Targeted signs or symptoms were related to dysfunction of the autonomic nervous system (parasympathetic or sympathetic or both). We took into account the clinically- manifested abnormalities during an oriented patient history taking and a meticulous physical examination assisted as needed by clinical tests.

The signs were (i) cardiovascular (orthostatic hypotension; leg edema; lengthening of the QT segment, and permanent tachycardia and ventricular arrhythmia on the electrocardiogram); (ii) digestive (dysphagia, retrosternal burn, gastroesophageal reflux; gastroparesis, nausea, vomiting, abdominal pain, bloating, abdominal distension, early satiety, urgent postprandial motor diarrhea, and constipation); (iii) urinary (perturbed perception of the need to urinate, dysuria, chronic urinary retention; urgent need to urinate, and imperious urination with leaks); (iv) genital signs: decreased vaginal secretions and anorgasmia in women, and retrograde ejaculation, erectile dysfunction in men; (v) pupillary motor abnormalities (abnormal photomotor reflex to light from a flashlight); (vi) damage to the sweat system (anhidrosis of the extremities of the lower limbs, dry skin, and hyperhidrosis, i.e., diffuse sweating).

Clinical tests were carried out to assess the sympathetic and parasympathetic systems. For the parasympathetic system, we measured changes in the heart rate (HR) during (i) deep breathing (the patient performed sequences of six (6) cycles per minute, then a maximum expiration over five (5) seconds); (ii) a Valsalva maneuver (15 s exhalation pressure). The maximum inspiration rate (I)/minimum expiration rate (E) was recorded. The normal value is I/E > 22 beats per minute (BPM) for patients under 60 years of age and I/E > 15 bpm for patients over 60 years of age. For the sympathetic system, blood pressure was measured in the orthostatic position at one, three, and five minutes to detect orthostatic hypotension. A decrease in blood pressure was considered significant when there was a drop of at least 20 mm Hg in the systolic pressure and/or 10 mm Hg in the diastolic pressure. A standard

electrocardiogram (EKG) assessment was performed during the consultation to determine the duration of the QT segment using an ECG ruler on Day 1, Day 2, and Day 3. The norms were < 0.43 s for men and < 0.45 s for women. We considered that the QT was lengthened or prolonged when it was > 0.45 s for men and > 0.47 s for women. Ventricular tachycardia was defined as a heart rate greater than 120 beats/min. The cutoffs for HbA1c for this work were less than 7% for type 2 diabetes and 7%-7.5% for type 1 diabetes. Autonomic neuropathy was presumed to be present when there were at least three (3) dysfunctions based on the clinical examination and tests [24].

**Dyslipidemia.** This was characterized by the presence of one or more of the following: high levels of total cholesterol and/or triglycerides, LDL cholesterol, or a low level of HDL. The cutoffs for lipoprotein were as follows:

Total cholesterol: normal value < 5.16 mmol/L, with a range of 5.16–6.16 mmol/L, and 6.20 mmol/L was considered high.

i. LDL cholesterol: normal value < 2.58 mmol/L, with a range 3.35–4.0 mmol/L, and 4.12–4.87 mmol/L was considered high.

ii. HDL cholesterol: a value was considered normal if < 1.0 mmol/L and high if ≥ 1.54 mmol/L.

iii. Triglycerides: normal value < 1.71 mmol/L and a value of ≥ 2.28 mmol/L was considered high.

**Obesity.** This was defined as patients who had a body mass index (BMI) ≥ 30 Kg/m$^2$.

**DN4 questionnaire.** This allows determination of the probability of neuropathic pain. It is divided into ten (10) "Yes" or "No" items to be checked, where Yes equals 1 point and No equals zero points. The final score is out of 10. When a patient scores ≥ 4/10, the test is positive, and it has a sensitivity of 82.9% and a specificity of 89.9% [25].

**Intensity of the pain.** A categorical scale was used that involved verbal inquiry of the patient to evaluate their pain subjectively by indicating one of the following five (5) categories: no pain; mild pain; moderate pain; intense pain; and extremely intense pain [25].

**The Michigan Neuropathy Screening Instrument (MNSI) test.** This is a diagnostic tool for DNP, and it has two parts (the patient's experience of the pain/discomfort and a strict clinical evaluation). A patient was considered to have DPN when their score was ≥ 7 on part one and ≥ 2.5 on part two of the test.

**The EuroQol 5 Dimensions (EQ-5D scales).** The EQ-5D scale provides a straightforward descriptive profile of the health status of a patient [26]. This instrument evaluates five (5) parameters (mobility; self-care; usual activities; pain/discomfort; and anxiety/depression). Each parameter is categorized as one of the following three (3) levels: no problem, some problems, and severe problems.

**Study design.** This work was carried out synchronously at two sites by a multidisciplinary team that included neurologists, diabetologists, and general practitioners. These two sites were the Gabriel Touré University Hospital Center of Bamako (GT-UHC), which is the 3$^{rd}$ reference healthcare facility in the country, designated as expert level in the Malian health pyramid, and the Center against Diabetes in Mali (CCD), which comprises the Malian association of patients with diabetes or the *Malian Association for the fight against Diabetes*. It is affiliated with the International Diabetes Federation. Its headquarters are in the capital city Bamako, with a focal point in all of the country's health districts. It represents a basic healthcare facility at the first level in the Malian health pyramid. The CCD sees an average of 12,660 patients with diabetes per month, while the GT-UHC averages 400 such patients every month. Therefore, these two health facilities were very substantial recruitment sites of patients with diabetes. We conducted a cross-sectional, descriptive study from May to November 2019. Our study

included volunteers who were 16 years of age or older who had type 1 or type 2 diabetes and who were able to understand and answer questions in French (the official language of the country) or in Bambara (the national language of the country) and who complied with a follow-up visit at the GT-UHC and/or the CCD. The sample size was estimated to be 252 using Kish's formula (n = $Z^2$ * (P*Q)/$i^2$) [27], with an estimated prevalence of diabetic polyneuropathy in Mali of 50% [28, 29], a 95% confidence interval yielding Z = 1.96, and accuracy i = 9%. The sampling was systematic based on the order of arrival of eligible patients at one of the two facilities, but randomized, i.e., only every fourth eligible patient was recruited.

## Data collection

We used three (3) questionnaires: one for the initial evaluation of patients with DPN, one for their current assessment, and another to survey the caregivers of patients with DPN. Data regarding the patients with DPN were collected during a structured face-to-face interview. The current questionnaire comprised the following seven (7) parts: (i) sociodemographic data, (ii) clinical and biological data related to diabetes, (iii) diagnosis of the DPN with the test (MNSI), (iv) DN4 to identify the neuropathic pain, (v) data related to their standard of living, (vi) the quality of life with the ED-5D scale, and (vii) the knowledge, practices, and therapeutic itinerary of patients with DPN. The questionnaire was pretested at a study site outside the CCD and the GT-UHC. This allowed us to shorten the interview. Prior to their recruitment into our study, the patients underwent testing for fasting glycemia, glycated hemoglobin (HbA1C), creatininemia levels, lipid levels (total cholesterol, HDL, LDL, and triglycerides), HIV serology, a complete blood count, serum electrolytes, syphilis serology (TPHA/VDRL), and uric acid levels. The data were collected in two stages: (i) a randomly selected patient completed an initial simplified questionnaire to determine the notion of chronic pain/discomfort suggestive of DPN with or without neuropathic pain (paresthesia, pain, falls, difficulty walking, balance disorder, numbness of the lower limbs, a burning sensation, other ailments), (ii) patients determined to be positive for DPN based on this screening would then complete our full study questionnaire. The translation of the questionnaire from French to Bambara and the back translation from Bambara to French were carried out at the National Directorate of Languages of Mali (DNAFLA).

## Survey of caregivers of patients with DPN

We surveyed 45 neurologists who were conveniently selected throughout Africa to document the issues of accessibility and availability of medications used to treat DPN. They filled out a structured, semi-open questionnaire, sent by email, inquiring about their prescription habits, the availability, accessibility, and the cost of prescription medications used to treat patients with DPN.

## Statistical analysis

The data were analyzed using Prism GraphPad version 8.0 software. Frequency tables and average values were produced. The $Chi^2$ test and Fisher's exact test were used to compare the proportions, with statistical significance at $p < 0.05$. A binary regression model with DPN as a dichotomous dependent variable (with/without DPN) was performed. The odds ratios (OR) with the 95% confidence intervals (CI) were used to measure the associations.

## Ethical considerations

The study was approved by the relevant authorities at the two study sites. Our study protocol was approved by the International Review Board (IRB) at the Faculty of Medicine and

Dentistry in Bamako, Mali. Our protocol has been validated by the ethics committee of the Faculty of Medicine and Pharmacy of Bamako under number N "2O2O / 209 / CE / FMOS-FAPH. We obtained informed consent from each eligible patient with DPN prior to their recruitment. We emphasized the aim and benefits of the study. The main benefit for our study participants was access to care of diabetic neuropathy with international standards. No monetary compensation was provided. The patients paid for their own blood analysis, which was part of their normal routine blood work. No additional examinations were performed at the patients' expense. Patients with DPN who either refused to participate or were not randomly selected were still entitled to their regular and routine care at both of the study sites without any prejudice.

## Results

### Sociodemographic and clinical data

Two hundred fifty-two (N = 252) patients with diabetes were randomly selected at the two health facilities (126 at the GT-UHC and 126 at the CCD) in Bamako. In total, the patients had diabetic polyneuropathy (DPN) in 69.8% (176/252) of cases, including 81.7% (103/126) from the GT-UHC and 57.9% (73/126) from the CCD. Type 2 diabetes was found in 83% (146/176) of the patients with DPN.

The patients were from Bamako in 85.8% (151/176) of cases, and 66% (116/176) of them were between 31 to 60 years of age. The sex ratio was approximately three females for every male. Our patients were housekeepers in 41.4% (73/176) of cases, with their highest level of education being primary school in 38.1% (67/176) or illiteracy (i.e., no formal schooling) in 34.7% (61/176). The patients did not earn a wage nor did they have a set salary in 66.5% (117/176) of cases. Only 19.9% (35/176) of the patients had an income higher than the minimum guaranteed interprofessional wage. **Table 1** summarizes the sociodemographic data.

The 176 patients included in the study had a positive score for the MNSI, i.e., the interview part of the MNSI (score $\geq$ 7) and the physical examination MNSI (score $\geq$ 2.5). The average delay in the onset of neurological signs was 6 ± 5 years. The patients had neuropathic pain in 86.9% (153/176) of cases based on the DN4. The delay in the onset of the neuropathic pain was relatively short, at less than two years in 19.9% (35/176) of cases. Neuropathic pain revealed diabetes in 17% (30/176) of the included cases.

The patients rated their pain as moderate or intense in 48.3% (85/176) of cases or as very intense in 37.5% (66/176). The pain was often perceived as either a burning sensation in 34.1% (60/176) (95% CI [27.4–41.3] of cases or as a tingling sensation in 24.4% (43/176) (**Table 2**).

### The neurological examination and risk factors associated with DPN

Based on the assessment with the DN4 tool, 86.9% (153/176) of the patients had neuropathic pain.

Thirty-two of the patients without DNP (n = 76) had signs of neuropathic pain, while some of the patients had other signs indicative of a mechanical or inflammatory cause of the pain.

The type of treatment in general and DPN were statistically associated (Fisher's exact test, p-value = .045, see **Table 2**).

The patients had impaired proprioceptive and exteroceptive sensations in 51.7% (91/176) of cases (95% CI [44.3–59.0]). Autonomic neuropathy (digestive, genital, sweat-related, and cardiovascular disorders) was found in 48.3% (85/176) of cases. Male patients reported erectile dysfunction in 69.6% (32/46) of cases. A varying degree of distal motor deficit was noted in the lower limbs in 27.8% (49/176) of cases (**Table 2**).

**Table 1. Sociodemographic data of patients with DPN.**

| Characteristics | Modalities | % DPN+ [95% CI] | DPN+ (n = 176) | DPN- (n = 76) | OR 95% CI | p |
|---|---|---|---|---|---|---|
| Study site | GT-UHC | 58.5 [51.1–65.6] | 103 | 32 | 1.9 [1.1–3.3] | 0.02 |
| | CCD | 41.5 [34.4–48.9] | 73 | 44 | 1 | - |
| Age ranges (in years) | 16–30 | 3.4 [1.4–6.9] | 6 | 52 | .002 [.0002-.0184] | .0001 |
| | 31–60 | 66.0 [58.7–72.6] | 116 | 23 | .09 [.01-.71] | .005 |
| | > 60 | 30.6 [24.2–37.8] | 54 | 1 | 1 | - |
| Gender | Male | 26.1 [20.0–33.0] | 46 | 29 | .57 [.32–1.02] | .05 |
| | Female | 73.9 [67.0–79.9] | 130 | 47 | 1 | - |
| Socio-professional ranking | Housewives | 41.5 [34.4–48.9] | 73 | 39 | .94 [.16–5.34] | .94 |
| | Government employees and retired officials | 28.9 [22.6–36.0] | 51 | 21 | 1.21 [.21–7.14] | .82 |
| | Blue collar job, merchants, farmers, and informal workers | 27.3 [21.1–34.2] | 48 | 14 | 1.71 [.28–10.36] | .55 |
| | Others * | 2.3 [0.7–5.4] | 4 | 2 | 1 | - |
| Residence | Bamako | 85.8 [80.0–90.4] | 151 | 63 | 1.25 [.60–2.59] | .55 |
| | Outside of Bamako | 14.2 [9.6–19.9] | 25 | 13 | 1 | - |
| Education level | Illiterate | 34.6 [27.9–41.9] | 61 | 41 | 1 | - |
| | Literate in Arabic or in the local language | 22.7 [16.9–29.4] | 40 | 21 | 1.28 [.66–2.48] | .46 |
| | Primary and secondary | 38.1 [31.1–45.4] | 67 | 6 | 7.51 [2.98–18.91] | .0001 |
| | Tertiary | 4.6 [2.1–8.4] | 8 | 8 | .67 [.23–1.93] | .46 |
| Income relative to the SMIG** | No fixed salary or self-employed | 66.5 [59.3–73.2] | 117 | 6 | 23.4 [9.19–59.61] | .0001 |
| | Salary below the SMIG | 13.6 [9.1–19.3] | 24 | 28 | 1.03 [.51–2.08] | .93 |
| | Salary above the SMIG | 19.9 [14.5–26.3] | 35 | 42 | 1 | - |
| Duration of the diabetes | < 1 year | 6.8 [3.7–11.3] | 12 | 39 | 1 | - |
| | 1–5 years | 30.1 [23.7–37.2] | 53 | 27 | 6.38 [2.88–14.14] | .0001 |
| | 5–10 years | 50.6 [43.2–57.9] | 89 | 7 | 41.32 [15.12–112.92] | .0001 |
| | > 10 years | 12.5 [8.2–18.0] | 22 | 3 | 23.89 [6.06–93.69] | .0001 |
| BMI | < 30 (not obese) | 41.5 [34.2–45.8] | 73 | 61 | 1 | - |
| | ≥ 30 (obese) | 58.5 [51.2–65.8] | 103 | 15 | 5.74 [3.03–10.88] | .0001 |
| Blood pressure | Low or optimal | 44.9 [37.5–52.2] | 79 | 55 | 1 | - |
| | High | 55.1 [47.8–62.5] | 97 | 21 | 3.22 [1.79–5.77] | .0001 |
| HbA1c | Optimal | 19.9 [14.0–25.8] | 35 | 52 | 1 | - |
| | Within normal limits or high | 80.1 [74.2–86.0] | 141 | 24 | 8.73 [4.75–16.05] | .0001 |
| Total cholesterol | Low or optimal | 44.3 [37.0–51.7] | 78 | 37 | 1 | - |
| | High | 55.7 [48.3–63.0] | 98 | 39 | 1.19 [.70–2.04] | .52 |
| HDL | Low or optimal | 57.4 [50.1–64.7] | 101 | 24 | 1 | - |
| | High | 42.6 [35.3–49.9] | 75 | 52 | .34 [.19-.61] | .0002 |
| LDL | Low or optimal | 38.1 [30.9–45.2] | 67 | 58 | 1 | - |
| | High | 61.9 [54.8–69.1] | 109 | 18 | 4.43 [2.38–8.23] | .0001 |
| Triglycerides | Low or optimal | 54.0 [46.6–61.3] | 95 | 53 | 1 | - |
| | High | 46.0 [38.7–53.4] | 81 | 23 | 1.96 [1.11–3.48] | .02 |

*Others = students, unemployed, retired without income

The patients with DPN were obese with and without dyslipidemia in 36.6% (65/176) (95% CI [27.9–41.9]) and 28.4% (50/176) (95% CI [22.1–35.4]) of cases, respectively. This association was found in 34.7% of the sample (61/176). The patients with DPN were heavy tobacco smokers in 3.40% (6/176) of cases (**Table 2**).

**Table 2. Clinical and therapeutic data of the patients with DPN.**

| Characteristics | Modalities | % DPN+ [95% CI] | DPN+ (n = 176) | DPN- (n = 76) | OR 95% CI | p |
|---|---|---|---|---|---|---|
| Intensity of the pain | Low | 14.2 [9.6–19.7] | 25 | 41 | 1 | - |
| | Moderate | 15.9 [11.1–21.9] | 28 | 27 | 1.70 [.8–3.5] | .15 |
| | Intense | 32.4 [25.8–39.6] | 57 | 6 | 15.6 [5.9–41.4] | .0001 |
| | Very intense | 37.5 [30.6–44.8] | 66 | 2 | 54.1 [12.2–240.6] | .0001 |
| Type of pain | Burning sensation | 34.1 [27.4–41.3] | 60 | 21 | 1 | - |
| | Tingling | 24.4 [18.5–31.2] | 43 | 2 | 7.5 [1.7–33.8] | .003 |
| | Paresthesia | 18.8 [13.5–23.0] | 33 | 51 | .2 [.1-.4] | .0001 |
| | Searing pain | 13.6 [9.1–19.3] | 24 | 1 | 8.4 [1.1–66.0] | .02 |
| | Numbness | 9.1 [5.5–14.0] | 16 | 1 | 5.6 [.7–44.8] | .07 |
| Foot examination | Normal (clean without a skin lesion) | 40.3 [33.1–47.6] | 71 | 63 | 1 | - |
| | Abnormal (dirty with visible skin lesions) | 59.7 [52.4–66.9] | 105 | 13 | 7.2 [3.7–14.0] | .0001 |
| Associated factors | Obesity | 28.4 [22.1–35.4] | 50 | 26 | .5 [.1–4.5] | .66 |
| | AHT | 5.1 [2.5–9.2] | 9 | 21 | .1 [.01–1.1] | .05 |
| | Dyslipidemia | 12.5 [8.2–18.0] | 22 | 17 | .3 [.03–3.2] | .63 |
| | Tobacco | 1.1 [0.2–3.7] | 2 | 3 | .2 [.01–2.8] | .52 |
| | Obesity + AHT | 7.4 [4.2–12.0] | 13 | 4 | .8 [.1–9.5] | .99 |
| | Obesity + Dyslipidemia | 34.7 [27.9–41.9] | 61 | 1 | 15.3 [.8–291.6] | .14 |
| | Obesity + Tobacco | 1.1 [0.2–3.7] | 2 | 2 | .3 [.01–4.7] | .52 |
| | AHT + dyslipidemia | 7.4 [4.2–12.0] | 13 | 1 | 3.3 [.2–64.6] | .46 |
| | Obesity + AHT+ Dyslipidemia + Tobacco | 2.3 [0.7–5.4] | 4 | 1 | 1 | - |
| Status of the neuropathic pain treatment | No treatment | 63.1 [55.7–69.9] | 111 | 61 | 1 | - |
| | Under treatment | 36.9 [30.0–44.2] | 65 | 15 | 2.4 [1.3–4.5] | .007 |
| Type of treatment in general* | Conventional medicine (CM) | 15.3 [10.6–21.2] | 27 | 21 | 1 | - |
| | CM and phytotherapy | 67.7 [60.4–74.2] | 119 | 41 | 2.3 [1.2–4.4] | .01 |
| | CM and Marabout | 12.5 [8.2–18.0] | 22 | 13 | 1.3 [.5–3.2] | .55 |
| | CM and scarification of the feet | 4.5 [2.1–8.4] | 8 | 1 | 6.2 [.7–53.7] | .13 |
| General therapeutic observance | No therapeutic interruption | 42.6 [35.5–50.0] | 75 | 42 | 1 | - |
| | Interrupted for financial reasons | 50.0 [42.6–57.4] | 88 | 20 | 2.5 [1.3–4.6] | .004 |
| | Interrupted due to unavailability | 1.7 [0.4–4.6] | 3 | 5 | .3 [.1–1.5] | .15 |
| | Interrupted for a traditional treatment | 5.7 [2.9–9.9] | 10 | 9 | .6 [.2–1.7] | .34 |

PD = Proprioception disorder, OED = Objective exteroception disorder

AN = Autonomous neuropathy, AHT = Arterial hypertension

## Therapeutic itinerary, attitudes, and practices of patients with DPN

Our study participants received no specific pain treatment in 63.1% (111/176) of cases, while in 36.9% (65/176) of cases they were treated with amitriptyline. The patients with DPN who were in pain were four times more likely to receive amitriptyline at the GT-UHC than those seen at the CCD, (odds ratio = 4.13, 95% CI = [2.05–8.32]) (**Fig 1.**).

The patients with DPN reported simultaneous use of conventional medicine along with traditional medicine in 84.7% (149/176) of cases and on its own in 15.3% (27/176). The patients with DPN were three times more likely to use both conventional and traditional medicine simultaneously at the CCD than those seen at the GT-UHC (odds ratio = 3.69, 95% CI = [2.05–8.32]) (**Fig 2**).

The most commonly used traditional practice was phytotherapy. Thus, most patients used leaves, bark, or roots of local plants as beverages or in fumigation. Another common

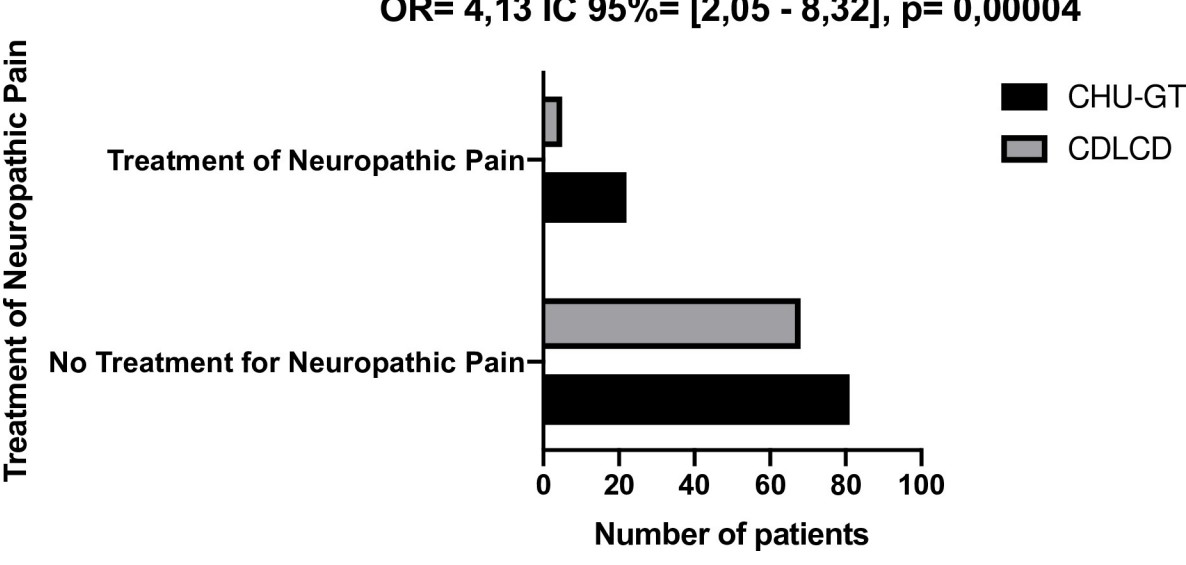

**Fig 1. Treatment status of patients in relation to neuropathic pain.**

traditional practice was the use of "*gris-gris*", a rope made by marabouts in which knots were made for incantations. Gris-gris were attached to the sore limbs, usually the thighs just above the knees (**Fig 3**). A less common traditional practice was scarification of the lower limbs (**Fig 4**).

### Knowledge, attitudes, and practices of patients with DPN

In 55.1% (97/176) of cases, the patients knew that their DPN was associated with diabetes. They reported irregular observance of the treatment for glycemic control during the follow-up

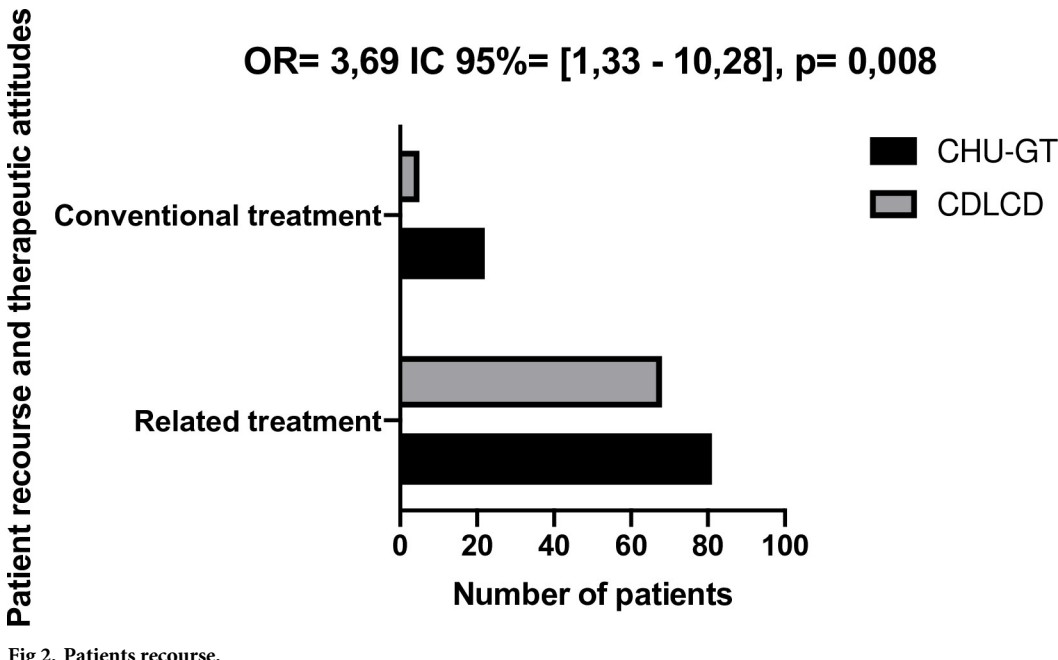

**Fig 2. Patients recourse.**

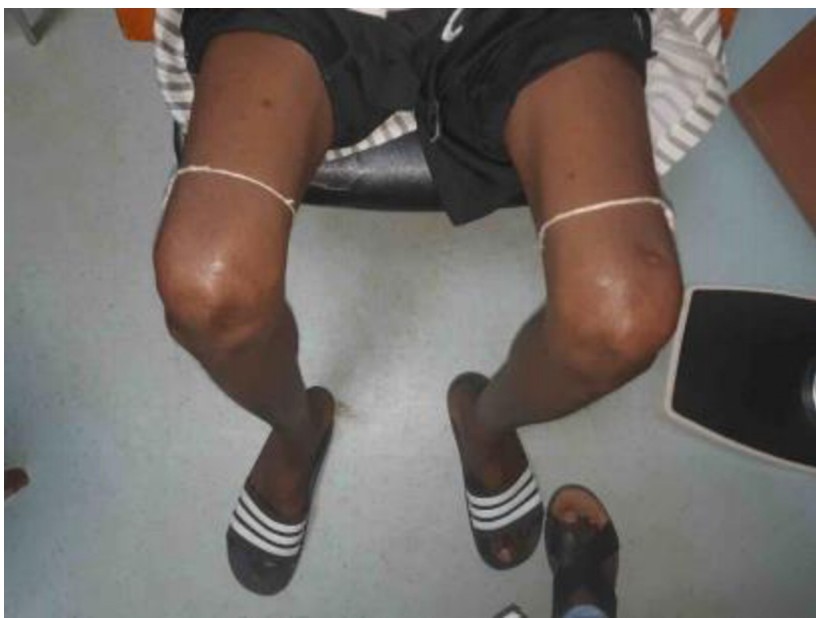

**Fig 3. *Gris-gris* from marabouts attached to the lower limbs of an adult male patient with DPN.**

period in 57.4% (101/176) of cases. The patients who were irregular observers either stated that they had financial issues in 88.1% (88/101) of cases, or they had decided to shift fully from conventional medicine to traditional medicine in 10% (10/101) of cases. In 23.29% (41/176) of cases, the patients did not observe their conventional treatment at all (**Table 2**).

## Quality of life of the patients with DPN

The five (5) parameters of the ED-5D scale on the quality of life of the patients with DPN were as follows (**Table 3**):

1. **Mobility:** The patients with DPN who were more than 60 years of age presented the most difficulty in regard to mobility, at 87.5% (21/24) p = $3.9^*10^-$. They were either confined or bedridden.

2. **Autonomy**: The patients with DPN reported difficulties in terms of grooming and dressing/undressing in 91/176 (51.7%) of cases. The patients who were 61 years of age or more had more DPN than those in the 16 to 30 and the 31 to 60 years of age brackets, p = 0.0001 and p = 0.005, respectively.

3. **Pain/discomfort:** The patients with DPN reported pain or discomfort in 39% (69/176) of cases. Approximately three-fourths, or 76.8% (53/69), of these patients were between 31–60 years of age.

4. **Anxiety/Depression:** The patients with DPN aged 31–60 year's old experienced severe anxiety or depression in 56% (14/25) of cases. The survey of the caregivers of the patients with DPN had an 82% (37/45) response rate. The treatments for DPN were clearly reported to be unaccessible and non-available to most patients. Table 4 summarizes the variation of drug costs in the treatment of neuropathic pain confronted with the national SMIG used in ten African countries (African Neurologist survey data).

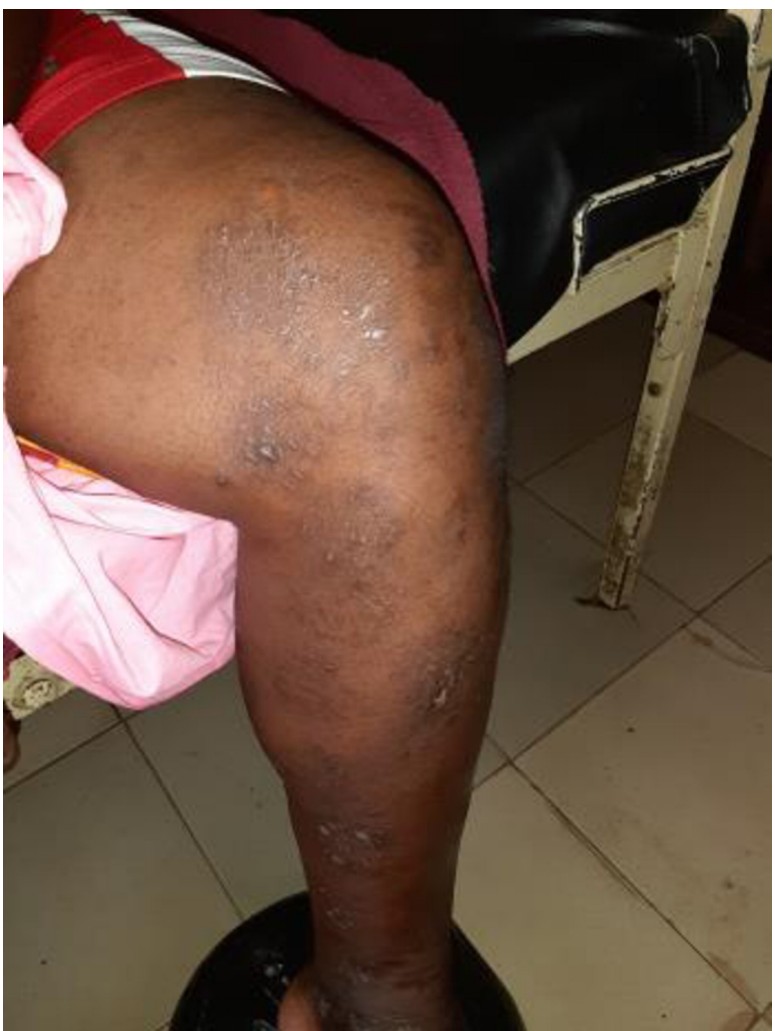

**Fig 4. Scarification of the lower limb in a patient with DPN.**

## Discussion

This work was carried out in the setting of our daily medical practice. It was inspired by the recommendations of the American Diabetes Association and the American Academy of Neurology consensus conference on DPN [28]. The DPN classification, from the above two scientific societies based on clinical aspects, makes it possible for clinicians in developing countries that lack sophisticated technical platforms to conduct clinical research work on this pathology with or without electrodiagnosis. However, in order for the clinical data to be robust, they recommended (i) the use of questionnaires or a validated interview technique and (ii) a rigorous neurological evaluation [28]. Our work fits these conditions perfectly. In addition, the 1988 recommendations were recently revised by the Toronto Diabetic Neuropathy Expert Group [12], which categorizes patients with DPN into four (4) groups based on clinical and paraclinical criteria: (i) possible DPN; (ii) probable DPN; (iii) confirmed DPN; and (iv) subclinical DPN. These authors recommend that groups 1, 2, and 3 are used for clinical practice and that groups 3 and 4 are used for research studies. In addition, a close correlation has been established between clinical data collected under rigorous conditions and neuropathological abnormalities in DPN [12].

**Table 3. The health profile of the population according to the ED-5D scale.**

| Health profile | | % DPN+ [95% CI] | DPN+ (n = 176) | DPN- (n = 76) | OR 95% CI | p |
|---|---|---|---|---|---|---|
| Mobility | No problem | 33.0 [26.0–39.9] | 58 | 68 | 1 | - |
| | Some problems | 53.4 [46.0–60.8] | 94 | 7 | 15.7 [6.8–36.6] | .0001 |
| | Restricted, bedridden | 13.6 [8.6–18.7] | 24 | 1 | 28.1 [3.7–214.4] | .0001 |
| Autonomy | No problem | 33.5 [26.5–40.5] | 59 | 70 | 1 | - |
| | Problems with autonomy | 51.7 [44.3–59.1] | 91 | 1 | 108.0 [14.6–798.5] | .0001 |
| | Incapable of grooming, getting dressed | 14.8 [9.5–20.0] | 26 | 0 | NA | - |
| Usual activities | No problems | 15.9 [10.5–21.3] | 28 | 57 | 1 | - |
| | Problems | 58.0 [50.7–65.2] | 102 | 19 | 10.9 [5.6–21.3] | .0001 |
| | Incapable | 26.1 [19.6–32.6] | 46 | 0 | NA | - |
| Pain/discomfort | Absence of pain/discomfort | 27.3 [20.7–33.9] | 48 | 44 | 1 | - |
| | Moderate pain/discomfort | 33.5 [26.5–40.5] | 59 | 30 | 2.1 [1.2–3.9] | .01 |
| | Extreme pain/discomfort | 39.0 [32.0–46.4] | 69 | 2 | 37.6 [8.7–162.7] | .0001 |
| Anxiety/Depression | Neither anxious nor depressed | 59.7 [52.4–66.9] | 105 | 39 | 1 | - |
| | Moderately anxious or depressed | 26.1 [19.6–32.6] | 46 | 35 | .5 [.3-.9] | .01 |
| | Extremely anxious or depressed | 14.2 [9.0–19.4] | 25 | 2 | 4.6 [1.1–20.5] | .02 |

## DPN prevalence in Mali is higher than the average prevalence in the literature

This cross-sectional study was conducted by a multidisciplinary team. This study was the first of its kind in French-speaking Western Africa to specifically address the various aspects of DPN and neuropathic pain. Data were collected from the GT-UHC, a leading reference healthcare facility at the third level, and from the CCD, a peripheral healthcare facility at the first level of the Malian health pyramid, which allowed us to obtain an overview on DPN in our context.

The prevalence of DPN was 69.8%, and type 2 diabetes was a common feature, occurring in 82.9% of cases. In line with most DPN prevalence studies from the Western countries, Truini et al. [30] in Italy reported a much lower DPN prevalence of 36%. In France, 4,400 adult patients with diabetes were monitored from 1947 to 1973. Half of these patients (50%)

**Table 4. Costs of drugs used in the medical treatment of neuropathic pain relative to the minimum national industrial wage in ten African countries (official sources).**

| Country | Drugs; Market price for one month of treatment (US dollars) | | | | | Minimum wage (US dollars) |
|---|---|---|---|---|---|---|
| | Neurontin® | Lyrica® 75 mg | Topalgic® 100 mg | Laroxyl® 25 mg | Tegretol® | |
| | Gabapentin 300 mg | Pregabalin 75 mg | Tramadol 100 mg | Amitriptyline 25 mg | Carbamazepine 400 mg | |
| Mali | 48 | 62 | 22.6 | 7.17 | 9.29 | 80.69 |
| Benin | 39.49 | 83.05 | 19.85 | 5.46 | 9.46 | 80.28 |
| Cameroon | - | 51.03 | 4.19 | 5.31 | 9.45 | 72.53 |
| Ivory Coast | 68.50 | 52.99 | 8.32 | 5.34 | 11.14 | 119.99 |
| Djibouti | 61 | 38 | 5 | 7.50 | 9.14 | 85.89 |
| Morocco | 67.34 | 116.60 | 21.70 | 16.47 | 8.95 | 336.67 |
| Niger | 41.28 | 52.38 | 16.48 | 5.85 | 11.60 | 60.08 |
| Senegal | 87.36 | 65.65 | 16.64 | 5.50 | 8.70 | 95.39 |
| Burkina Faso | 45.80 | 44.27 | 6.10 | 4.58 | 10.68 | 64.42 |
| Togo | 53.78 | 48.12 | 4.26 | 6.30 | 10.20 | 69.99 |

developed a peripheral neuropathy by the end of the follow-up period [31]. Another European study included both type 1 and type 2 diabetes patients. The authors reported an overall prevalence of 43% (50.8% type 2 diabetes and 25.6% type 1 diabetes) [11]. In Denmark, the prevalence of DPN was reported to be 18% in patients recently diagnosed (i.e., less than five years) with diabetes [32]. In the USA, a review of the literature over a 10-year period that included 321 articles reported a DPN prevalence of 26.8% [14].

A number of studies from African countries have reported results similar to those in Western countries. In South Africa, a study carried out at both public and private healthcare centers involving 1,046 patients with type 1 or type 2 diabetes were monitored by 50 diabetes treatment centers, yielding a DPN prevalence of 30.3% [15]. In Morocco, the prevalence of DPN has been reported to be 15% [33]. In Egypt, the overall prevalence of DPN in patients with diabetes has been reported to be 29.3% [34]. In Ethiopia, a study on the quality of life of 220 patients with diabetes has reported a prevalence of DPN of 78.9% [4].

In practice, the disparity between the prevalences of DPN across countries is considerable. Overall, the data showed that approximately a third of patients with diabetes developed DPN with or without pain [35].

The relatively high prevalence of DPN in our cohort compared to the prevalences reported in the above-mentioned studies could be explained by several factors, particularly the performance of the diagnostic tools (MNSI, DN4), combined with examination of patients by a multidisciplinary (neurologist, diabetologist) and experienced research team. Furthermore, it has been well-established that the MNSI is a straightforward and non-invasive tool validated in the context of DPN diagnosis. Other risk factors such as the relatively high average age of our patients, a long duration of the disease, and poor glycemic control may also explain the high prevalence of DPN in our study.

The MNSI has been shown to be effective compared to other more invasive tools such as electroneuromyography, sensory system tests, cutaneous biopsies, and Sudoscan™ tests, which are in fact largely unaccessible in Mali. The MNSI, when optimally used in association with a thorough clinical examination by an experienced clinician, has an increased sensitivity for diagnosing DPN [36, 37]. We found a female predominance and a higher frequency of type 2 diabetes, which is in line with what has been reported in the literature [35, 38].

## The risk factors for DPN in Africa in general and Mali in particular differ from those found elsewhere

In contrast to the Western countries, the main risk factors associated with DPN in Mali were obesity (28.4%) and dyslipidemia (12.5%). In the Western world, excessive consumption of alcohol and tobacco are known risk factors for DPN [33, 34]. Alcohol and tobacco have, however, very rarely been reported as associated factors in African studies. Indeed, in a study of 300 patients with diabetes in Morocco, alcohol and tobacco were not mentioned as factors associated with the occurrence of DPN [33]. In our study, only 4.5% of the patients smoked tobacco. Female patients formed the bulk of our sample, and for sociocultural reasons women in Mali generally do not consume either tobacco or alcohol.

Obesity was found to be the main factor in 73.9% of our patients. A significant increase in the average BMI has been observed in Africa over the past 35 years [39]. Furthermore, a steeper curve was noted in African women relative to trends worldwide. In all African countries, there is a strong positive association between the average BMI and the prevalence of diabetes, irrespective of gender [39].

Other than the recruitment bias, discordance in the prevalence between different areas could be due to a racial or genetic difference in susceptibility. Indeed, racial differences have

been reported in DPN [35]. Other studies have obtained evidence of a genetic susceptibility, as they showed a tight link between painful DPN and certain hereditary channelopathies such as mutation of the *SCN9A* gene, which codes for the sodium channel Nav1.7 [40, 41]. This genetic disparity underscores the need for clinical as well as genetic studies in Africa to further explore this issue.

The pathogenesis of DPN remains unclear, although its origin appears to be multifactorial [42]. The factors thought to be involved are intricate and comprise: (i) direct axonal attack of metabolic origin due to hyperglycemia, (ii) ischemic axonopathy secondary to microangiopathy, (iii) an autoimmune or chronic inflammatory phenomenon, and (iv) a genetic predisposition.

Furthermore, oxidative stress, hyperglycemia, dyslipidemia, and resistance to insulin can give rise to mitochondrial dysfunction [43]. Affliction of the non-myelinated C-fibers is responsible for the pain and certain subjective sensory manifestations (allodynia, paresthesia) [44]. At the pathological level, segmental axonal demyelination has been noted, followed by full axonal degeneration [45]. Randomized clinical trials have shown that glycemic control can slow the progression of peripheral neuropathy in DPN [46].

## DPN has major socio-economic consequences

DPN is predominant in patients who are 30–60 years of age, with predictable socio-economic consequences for them and society. This situation could become a major issue in Africa in light of the increasing number of patients with diabetes. In addition to being a pressing health issue, comorbidity of diabetes and HIV/AIDS could restrain development because both are prevalent among young adults and adolescents [47].

The low socio-economic level of patients with diabetes has been widely reported in the African medical literature [47, 48]. Most patients in our study had a low socio-economic status. They had no or a minimal salary, below the interprofessional guaranteed minimum wage in Mali (SMIG, 80 USD). In Ethiopia, the average monthly income of patients with diabetes was 58.6 USD [4]. In Zambia, individuals with type 2 diabetes reported psychosocial difficulties, a lack of information, and difficulty meeting basic needs such as food and transportation [48].

Most of the patients with DPN in our study had no or little formal education. The low level of education could negatively influence the quality of life of such patients. A good level of education would promote compliance with the treatment and result in increased earnings [48, 49].

Our patients indicated irregularities in regard to being able to follow the maintenance treatment in 57.4% of cases, and 23.3% of the patients reported a complete interruption of their treatment. Non-compliance with treatment is well known in patients with diabetes in Africa [50, 51].

In our study, 86.9% of the patients afflicted with DPN exhibited neuropathic pain. It has been reported that four out of five patients with diabetes experience neuropathic pain at some point in the course of the disease [52]. In our work, the DPN was not painful in 13.1% of the patients. The identification of patients with DPN without pain remains a clinical and diagnostic challenge as they often go unnoticed. These patients can potentially develop irreversible neurological damage that leads to amputation of the affected limbs [53].

The delay in the occurrence of the neuropathic pain was short (less than two years) in 19.9% of the patients. Diabetes was diagnosed due to the onset of neuropathic pain in 17.04% of the patients. Our results are in line with what has been reported in the literature in this regard. The early occurrence (within two years) of the neuropathic pain in newly diagnosed patients with type 2 diabetes has also recently been reported elsewhere [54].

### In the context of scarceness of electrophysiologic equipment in most African neurology departments, the MNSI is a simple yet effective tool for a positive diagnosis of DPN

We opted to use the MNSI to diagnose DPN. This tool is based entirely on the typically encountered symptoms in DPN [55]. It has a low sensitivity (40%) and a high specificity (92%), which makes it important to combine its use with a thorough clinical examination by an expert [56].

Thus, there are five neurological syndromes that are typically linked with diabetes: symmetrical polyneuropathy (the most frequent), autonomic neuropathy, polyradiculopathy, small-fiber neuropathies, and mononeuropathies [57, 58]. At the clinical level, sensory signs and autonomic neuropathy are often the inaugural symptoms [59, 60]. The sensory symptoms often recap the clinical picture. In our study, nearly all of the patients with DPN exhibited exteroceptive or proprioceptive disorders. Typically, motor manifestations occur at the advanced stages of the disease [12]. Only 27.8% of the patients were examined for motor disorders. Autonomic neuropathy was present in 48.3%. Indeed, autonomic neuropathy is one of the most common and serious complications of diabetes. Cardiac, digestive, sweat gland, and erectile dysfunction remain the most frequent manifestations [39]. The presence of autonomic neuropathy, particularly the cardiac manifestation, has been correlated with an elevated mortality rate [61]. In our study, 69.6% of the male patients with DPN reported erectile dysfunction versus 50% in the study by Mwila et al. in Zambia [48]. A Nigerian study reported known inadequacies in the treatment of DPN in Africa [61].

### EQ-5D is a suitable tool to evaluate the quality of life of patients

The analysis of the five parameters of the EQ-5D scale (mobility, autonomy, usual activities, discomfort/pain, and depression/anxiety) revealed a negative impact of DPN on the quality of life of the patients, especially in the elderly ones. The EQ-5D scale is a straightforward tool at the cognitive level. It only takes a few minutes to administer and is widely used to evaluate the various parameters of the quality of life, both in the general population and in clinical research. However, its performance appears to depend on the sociocultural realities of the studied population, as the concept of quality of life is not universal [62].

In Africa, studies in Zimbabwe and in Nigeria have validated this scale as a reliable tool for evaluation of the quality of life, even in patients with a low level of education [63, 64]. Of all the complications of diabetes, DPN with or without pain has the greatest association with anxiety and depression. In practice, the physical and mental components of the quality of life are significantly altered in patients with DPN. In the literature, anxiety, depression, and sleep disorders have been noted in approximately 43% of patients with DPN [19]. This situation warrants implementation of strategies even in the absence of pain [19].

### The use of traditional medicine is a reality. It should, therefore, be integrated into the conception of adopted health strategies

DPN in its conventional form is irreversible. The treatment hence aims to prevent progression of the disease and its associated complications [65]. The three main principles of this treatment are based on three entities: glycemic control, foot care, and management of the pain.

Management of neuropathic pain is currently lacking. Indeed, only 36.9% of the patients had a treatment targeting the neuropathic pain, based on amitriptyline alone. This drug is the only one prescribed as the first-line therapeutic arsenal in DPN and validated in the guidelines [66]. This situation shows the lack of training of caregivers in regard to the treatment of neuropathic pain in patients with DPN.

Furthermore, this deficiency in the treatment of neuropathic pain appears to be more pronounced in the CCD than in the GT-UHC, thus pointing out the need to train caregivers at the peripheral healthcare facilities. This therapeutic gap in the treatment of neuropathic pain appears to be a general phenomenon as it has also been reported previously in several European and African studies [52, 61]. We believe that, in our context, other than the training of caregivers, this gap is probably linked to the inaccessibility (high cost) and the non-availability of certain first-line drugs that have been shown to be effective in neuropathic pain.

In practice, in light of the substantial impact of DPN on the health, the quality of life of the patients, and the cost of the care, the American Diabetes Association currently recommends screening for DPN in all patients with diabetes at the time of their diagnosis along with annual visits for monitoring [67].

Most patients with DPN concomitantly used traditional and conventional medicines. In a work investigating the use of traditional medicine among patients with diabetes in three African countries, only 28% of the traditional healers in Mali, 20% of those in Mozambique, and 16% of those in Zambia referred their patients to a hospital to manage complications from diabetes [68]. This finding highlights the need to establish a setting for innovative exchange between traditional therapists and the conventional health system in order to achieve optimal treatment of diabetes and its complications. Moreover, the WHO estimates that 80% of the population of sub-Saharan Africa uses traditional medicine [69]. Furthermore, certain plants used in traditional medicine have anti-diabetic potential [70, 71].

## Limitations of the study

The limitations are inherent to the problem of adapting the tools of the survey. We believe that adapting these tools validated in other areas of the world remains an essential prerequisite for avoiding distortion of the information collected from the general population. We have also noted difficulties in the use of certain files as, in keeping with the health policies of Mali, the cost of performing additional examinations was borne entirely by our patients, with the risk of results that are often not robust. Lastly, we have often had difficulty explaining the quality of life, as this concept is not a universally held notion. Tailoring this concept to the African socio-cultural realities could be important for further studies. Markers based on electrodiagnosis, sensory system testing, nerve biopsy, skin biopsy, corneal confocal microscopy, pharmacological tests, skin vasomotor reflexes, and pupillometry are not available in Mali. We, therefore, did not use any such markers in this study. In addition, the data were collected over a relatively short period of time that was independent of us for two main reasons: (i) completion of the study questionnaire took approximately 40 minutes, which was too long for some of the patients and even for a number of physicians from the CCD because they were part-time workers or often volunteers in this center (ii) MD theses had time constraints since the residents in neurology had to be graded by the end of their fourth year.

## Conclusion

Our study confirms the high prevalence of DPN with or without pain in Mali. The diagnosis of DPN can be ensured in the African context by straightforward, inexpensive, and easy to use screening tools (MNSI, DN4). Screening and early treatment of patients at risk should allow the burden of DPN to be reduced. Training of caregivers remains a priority. The best approach to managing diabetes is to consider the sociocultural realities of patients with diabetes, as has been suggested in the literature [72].

## Supporting information

**S1 Data.**
(XLSX)

## Author Contributions

**Conceptualization:** Youssoufa Maiga, Cheick Oumar Guinto, Hamar A. Traoré, Julien Nizard.

**Data curation:** Youssoufa Maiga, Salimata Diallo, Seybou H. Diallo, Saliou Mahamadou, Awa Coulibaly, Mariam Daou, Zoumana Traoré, Mohamed Albakaye, Madani Ouologem, Adama S. Sissoko, Souleymane Papa Coulibaly.

**Funding acquisition:** Youssoufa Maiga.

**Investigation:** Youssoufa Maiga, Zoumana Traoré, Djeneba Sow Sylla, Mohamed Albakaye, Madani Ouologem, Adama S. Sissoko, Souleymane Papa Coulibaly.

**Methodology:** Youssoufa Maiga, Fatoumata dite Nènè Konipo, Oumar Sangho, Modibo Sangaré.

**Project administration:** Youssoufa Maiga.

**Software:** Youssoufa Maiga.

**Supervision:** Youssoufa Maiga, Hamar A. Traoré, Julien Nizard.

**Validation:** Yann Péréon, Cheick Oumar Guinto, Philippe Damier, Nadine Attal, Julien Nizard.

**Visualization:** Youssoufa Maiga, Yann Péréon, Cheick Oumar Guinto, Philippe Damier, Nadine Attal.

**Writing – original draft:** Youssoufa Maiga, Hamar A. Traoré.

**Writing – review & editing:** Youssoufa Maiga, Yann Péréon, Bernard Giumelli, Hamar A. Traoré, Philippe Damier, Nadine Attal, Julien Nizard.

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
