## [Decision Letter · Decision Letter 0]

29 Jul 2020

PONE-D-20-18728

Diabetic polyneuropathy with/out Neuropathic pain in Mali: a cross-sectional study in two reference Diabetes treatment centers in Bamako (Mali), Western Africa

PLOS ONE

Dear Dr. MAIGA,

Thank you for submitting your manuscript to PLOS ONE. After careful consideration, we feel that it has merit but does not fully meet PLOS ONE’s publication criteria as it currently stands. Therefore, we invite you to submit a revised version of the manuscript that addresses the points raised during the review process.

Please address all the major issues raised by both reviewers.

We look forward to receiving your revised manuscript.

Kind regards,

Rayaz A Malik, MBChB, PhD

Academic Editor

PLOS ONE

Additional Editor Comments:

Whilst your data is potentially useful and informative, the paper is let down by inadequate analysis and presentation.

The queries need to be addressed fully.

Journal Requirements:

3. Please amend the manuscript submission data (via Edit Submission) to include author Souleymane Papa Coulibaly.

Reviewers' comments:

Reviewer's Responses to Questions

**Comments to the Author**

1. Is the manuscript technically sound, and do the data support the conclusions?

Reviewer #1: Partly

Reviewer #2: Partly

2. Has the statistical analysis been performed appropriately and rigorously? 

Reviewer #1: No

Reviewer #2: No

3. Have the authors made all data underlying the findings in their manuscript fully available?

Reviewer #1: No

Reviewer #2: No

4. Is the manuscript presented in an intelligible fashion and written in standard English?

Reviewer #1: No

Reviewer #2: No

5. Review Comments to the Author

Reviewer #1: This is the second study to determine the prevalence of diabetic peripheral neuropathy (DPN) in diabetes (T1D & T2D) in secondary health care in Bamako, Mali. The study recruited 252 patients with diabetes and the prevalence of DPN was 176/252 (69.8%), which is relatively high compared to other countries. Although the paper is not of a publishable standard due to low sample size, poorly written, incorrect statistical analysis and misdiagnosis of autonomic neuropathy, I would recommend it for major revision on the basis of novelty for including the level of income and education which may have an important association with DPN and sufficient data have been collected.

The recruitment lasted for 8 months. I would recommend to the authors to recruit more patients given that the Center for Combating Diabetes (CCD) has an average of 12660 patients with diabetes per month. If the circumstances do not allow this add this as a study limitation.

Correct the power analysis. The previously reported prevalence of DPN by Traore et al. that you used for your sample size calculation was not 50% but 67.7%. So you will have to state that with a population estimate of 12660 in CCD, an estimated prevalence of DPN of 67.7% and margin of error of 6% the minimum sample is 230. It is not recommended to exceed 5% margin of error but with 5% margin of error the minimum sample size is 328, which is higher than your sample size (n=252). I want to emphasize that this power analysis is only to determine the prevalence of DPN. It is not enough to assess for association between risk factors like obesity, dyslipidemia with DPN. Mention this as a study limitation.

The paper is poorly written, not only grammatically but also there are incomplete sentences (i.e. aimed to determine the prevalence to describe) and inappropriate use of words or terms (i.e. treatments remain “pregnant” in Africa/ diabetics patients /one for the initial evaluation “on” for present evaluation). Avoid using the term diabetic, instead use patient with diabetes. The writing needs significant improvement.

If the combination of obesity and dyslipidemia was associated with DPN, please show in the table in frequency as n (%) of those with obesity and dyslipidemia in the group with/out DPN and provide the p value using chi-square analysis. Avoid repeating the percentage twice in the tables. Why is the percentage of combination of obesity and dyslipidemia 34.7 and 36.6. It should be 36.6%.

You show the percentage of different level of education, income, gender in Table 1 but you have not determined what is their association with DPN. Do a chi-square analysis and include the frequency between those with/out DPN and the p value.

Was there no association between those been treated with conventional, unconventional (i.e. Voodoo amulet) treatment and no treatment with DPN?

Were any of the 5 parameters from the quality of life assessment associated with DPN? The reporting of the percentage of people with difficulty with grooming and getting dressed, mobility, anxiety, day-to-day activities are not related to the objective of the study if you have not assessed for their association with DPN. Do a chi-square analysis or remove this data if the numbers between those with/out DPN are not enough. Without this you can’t state in your conclusion that “DPN has a negative impact on the quality of life, independently of the pain”.

I can’t emphasize on how much important is that you keep the analysis focused and simple, remove irrelevant information, shorten the results and discussion and avoid errors. Remove Table 3 because this is not a review paper. Although you do not need to repeat in the results everything written in the Tables you need to be consistent. In the results 38.1% have done primary school but in the table 38.1% are from primary and secondary school. Similar thing with the income level, check the discrepancy about the 66.5% in the results and Table 1.

Autonomic neuropathy cannot be defined by one sign only (i.e. urinary retention, erectile disorder or cuts or sores on the feet). It is best to leave this out.

In the methods you mentioned that HbA1c obtained, HbA1c was not included in the analysis. Why is that? If you have enough HbA1c data you should include it.

If the Center for Combating Diabetes (CCD) is composed of hospitals or centers inside or outside Bamako mention this in the methods. It is recommended that you include the study design and ethical approval before the assessments.

In the authorship, Nadine ATTAL has no affiliation and none of the authors are affiliated with Federal Pain Palliative Care and Support.

Reviewer #2: The study examined the prevalence of painful and painless diabetic peripheral neuropathy (DPN) in Mali and fills an important gap in literature in this country. Moreover, it highlights the lack of treatment of a very large proportion of patients with painful Diabetic Neuropathy and the impact of that on quality of life . As such, the findings are interesting , however, there are major issues with the methodology , missing data , presentation of the data and statistical analyses methods which may have impacted on the conclusions drawn by the authors, which I'll detail in the coming points :

1 The baseline characteristics data is very poorly presented and a lot of important data is missing -The mean age, weight, BMI, duration of diabetes, HbA1c, BP, and details of lipids panel values are lacking from the baseline characteristics table(1) and Table 11. These are important risk factors that need clear presentation in the main baseline characteristics table. It would be very important to compare between those who were diagnosed with DPN vs those who weren't . The same comparisons need to be made between those who had painful DPN and those who do not have the condition.

2-The association of risk factors with DPN in Table 11 is not very informative . while obesity was clearly defined with BMI > 30, dyslipidemia has not been defiend in the methods section. The way of obtaining the odd ratios was not cleary defined in the statistical methods. Ideally, Binary regression model with DPN as a dichotomous dependent variable and feeding in all risk factors in the model including gender, duration of diabetes, HbA1c, age, plus the other variables included in table 1 and then show the independent predictors of DPN and painful DPN with the odd ratios and P values.

3-The authors did not mention any information about the prevalence of foot ulceration and amputation in this sample. This is important to show the magnitude of the problem of the end product of this neglected complication of diabetes.

4-While the diagnosis of DPN is a straightforward clinical one and MNSI and DN4 are very robust, its a diagnosis of exclusion . Has B12, folate deficiency and other causes of neuroapathy been excluded ? Its important to mention this in the methodology to acknowledge that in limitations section if not done.

5-The authors attributed the high prevalence of DPN in this study to the performance of the diagnostic tools used together with the expertise of the team, however, other risk factors such as old age, longer duration of diabetes and poor glycemic control may be contributing factors and need to be mentioned in the discussion .

6- The only medication mentioned by the authors for painful neuropathy is Amitriptyline, how about the use of other agents mentioned in Table IV? any combinations ? Is it a cost issue alone or individual physician treatment approaches ?

7-The English language needs major editing for grammar and spelling mistakes and sentence formation , ideally by a native English-speaker .

6. PLOS authors have the option to publish the peer review history of their article (what does this mean?). If published, this will include your full peer review and any attached files.

Reviewer #1: **Yes: **Georgios Ponirakis

Reviewer #2: No

---

## [Author Response · Author response to Decision Letter 0]

7 Oct 2020

Memo of Responses to the Reviewers’ comments

From: Professor Youssoufa MAIGA

Corresponding author

 Bamako, September 5th, 2020

To: The PLOS One editorial office

Thank you for your thorough review with many very useful comments. Below are our responses (highlighted in green) to the reviewers’ comments. 

Foreword: 

The conclusions of the American Diabetes Association and the American Academy of Neurology consensus conference (Report and recommendations of the San Antonio Conference on diabetic neuropathy. Diabetes 37: 1000–1004, 1988), served as the basis for this work.

The following paragraph has been added on page 16 of the revised manuscript.

“This work was carried out in the setting of our daily clinical medical practice. It was inspired by the recommendations of the American Diabetes Association and the American Academy of Neurology consensus conference on DPN [28]. The DPN classification, from the above two scientific societies based on clinical aspects, makes it possible for clinicians in developing countries that lack sophisticated technical platforms to conduct clinical research work on this pathology with or without electrodiagnosis. However, in order for the clinical data to be robust, they recommended (i) the use of questionnaires or a validated interview technique and (ii) a rigorous neurological evaluation [28]. Our work fits these conditions perfectly. In addition, the 1988 recommendations were recently revised by the Toronto Diabetic Neuropathy Expert Group [12], which categorizes patients with DPN into four (4) groups based on clinical and paraclinical criteria: (i) possible DPN; (ii) probable DPN; (iii) confirmed DPN; and (iv) subclinical DPN. These authors recommend that groups 1, 2, and 3 are used for clinical practice and that groups 3 and 4 are used for research studies. In addition, a close correlation has been established between clinical data collected under rigorous conditions and neuropathological abnormalities in DPN [12].” 

Journal Requirements:

Comment 1. Please ensure that your manuscript meets PLOS ONE's style requirements, including those for file naming. The PLOS ONE style templates can be found at

 Response 1: The journal requirements have been taken into account in the revised version of the manuscript. 

Comment 2. PLOS requires an ORCID iD for the corresponding author in Editorial Manager on papers submitted after December 6th, 2016. Please ensure that you have an ORCID iD and that it is validated in Editorial Manager. To do this, go to ‘Update my Information’ (in the upper left-hand corner of the main menu), and click on the Fetch/Validate link next to the ORCID field. This will take you to the ORCID site and allow you to create a new iD or authenticate a pre-existing iD in Editorial Manager. Please see the following video for instructions on linking an ORCID iD to your Editorial Manager account: https://www.youtube.com/watch?v=_xcclfuvtxQ

Response 2: My ORCID iD is 0000-0003-4102-8328

Comment 3. Please amend the manuscript submission data (via Edit Submission) to include author Souleymane Papa Coulibaly.

 Response 3: Information on the co-author Souleymane Papa Coulibaly has been added. 

Comment 4. Has the statistical analysis been performed appropriately and rigorously? 

Response 4: We have included additional statistical analyses in order to be more rigorous. 

Comment 5. Have the authors made all data underlying the findings in their manuscript fully available?

Response 5: All of the data will be made available in a public repository. 

Comment 6. Is the manuscript presented in an intelligible fashion and written in Standard English?

Response 6: This new version of the manuscript has been edited by a native English speaker to correct typographical and grammatical errors. 

Reviewer #1:

This is the second study to determine the prevalence of diabetic peripheral neuropathy (DPN) in diabetes (T1D & T2D) in secondary health care in Bamako, Mali. The study recruited 252 patients with diabetes and the prevalence of DPN was 176/252 (69.8%), which is relatively high compared to other countries. Although the paper is not of a publishable standard due to low sample size, poorly written, incorrect statistical analysis and misdiagnosis of autonomic neuropathy, I would recommend it for major revision on the basis of novelty for including the level of income and education which may have an important association with DPN and sufficient data have been collected.

This prevalence of approximately 50% DPN in diabetic patients is what has been reported previously in Mali (Traoré 2014) and generally in the literature (Bouhassira et al. PLoS ONE 2013). The estimated prevalence of 50% of diabetic polyneuropathy was used to calculate the size of our sample, and a sample of 252 patients largely meets our requirements.

We believe that the prevalence of 50% found in Mali in a health district (first reference level after the peripheral facilities represented by community healthcare centers and dispensaries), seemed to us to be relatively high before the start of our study. Our work carried out at a 3rd level of the healthcare pyramid (the Gabriel Touré UHC), which recruits long-standing diabetic patients; and in a facility entirely dedicated to diabetic patients could not have a lower prevalence than that found in a health district. In addition, it has been shown that up to 50% of diabetic peripheral neuropathies can be asymptomatic, the demonstration of which requires rigorous neurological evidence. [Rodica Pop-Busui Care 2017; 40: 136–154]

Comment 1. What is the added value that our work brings as compared to the previous work carried out in Malian and in other African studies?

Response 1: This work provides added value compared to previous work carried out on the issue in Mali and in sub-Saharan Africa. This added value is as follows: (i) the work was carried out at two facilities dedicated to providing care for diabetic patients (a peripheral associative healthcare facility and a 3rd level healthcare facility in the Malian healthcare pyramid system); (ii) it involved a multidisciplinary team comprising neurologists, diabetologists, psychiatrists, and epidemiologists; (iii) it used validated, simple, and inexpensive tools that can readily be translated into local languages (MNSI, DN4, ED-5D); (iv) it took into account the sociocultural and economic aspects of DPN management. To our knowledge, this work is the first to integrate all of these important aspects into the African context. In addition, it has been reported that there is a close correlation between clinical data collected under rigorous conditions and the neuropathological abnormalities of the DPN. 

The definition of Diabetic Neuropathy has been revised in the manuscript.

The following paragraphs have been added on pages 5-7 of the revised manuscript.

“The screening for autonomic neuropathy (AN) was purely clinical based on an interview, physical examination, and available non-invasive clinical tests, as defined by the American Diabetes Association [13]. Targeted signs or symptoms were related to dysfunction of the autonomic nervous system (parasympathetic or sympathetic or both). We took into account the clinically-manifested abnormalities during an oriented patient history taking and a meticulous physical examination assisted as needed by clinical tests.

The signs were (i) cardiovascular (orthostatic hypotension; leg edema; lengthening of the QT segment, and permanent tachycardia and ventricular arrhythmia on the electrocardiogram); (ii) digestive (dysphagia, retrosternal burn, gastroesophageal reflux; gastroparesis, nausea, vomiting, abdominal pain, bloating, abdominal distension, early satiety, urgent postprandial motor diarrhea, and constipation); (iii) urinary (perturbed perception of the need to urinate, dysuria, chronic urinary retention; urgent need to urinate, and imperious urination with leaks); (iv) genital signs: decreased vaginal secretions and anorgasmia in women, and retrograde ejaculation, erectile dysfunction in men; (v) pupillary motor abnormalities (abnormal photomotor reflex to light from a flashlight); (vi) damage to the sweat system (anhidrosis of the extremities of the lower limbs, dry skin, and hyperhidrosis, i.e., diffuse sweating).

Clinical tests were carried out to assess the sympathetic and parasympathetic systems. For the parasympathetic system, we measured changes in the heart rate (HR) during (i) deep breathing (the patient performed sequences of six (6) cycles per minute, then a maximum expiration over five (5) seconds); (ii) a Valsalva maneuver (15 s exhalation pressure). The maximum inspiration rate (I)/minimum expiration rate (E) was recorded. The normal value is I/E > 22 beats per minute (BPM) for patients under 60 years of age and I/E > 15 bpm for patients over 60 years of age.

For the sympathetic system, blood pressure was measured in the orthostatic position at one, three, and five minutes to detect orthostatic hypotension. A decrease in blood pressure was considered significant when there was a drop of at least 20 mm Hg in the systolic pressure and/or 10 mm Hg in the diastolic pressure.

A standard electrocardiogram (EKG) assessment was performed during the consultation to determine the duration of the QT segment using an ECG ruler on Day 1, Day 2, and Day 3. The norms were < 0.43 s for men and < 0.45 s for women. We considered that the QT was lengthened or prolonged when it was > 0.45 s for men and > 0.47 s for women. Ventricular tachycardia was defined as a heart rate greater than 120 beats/min. The cutoffs for HbA1c for this work were less than 7% for type 2 diabetes and 7%-7.5% for type 1 diabetes. Autonomic neuropathy was presumed to be present when there were at least three (3) dysfunctions based on the clinical examination and tests [24].

Dyslipidemia: this was characterized by the presence of one or more of the following: high levels of total cholesterol and/or triglycerides, LDL cholesterol, or a low level of HDL.

The cutoffs for lipoprotein were as follows: 

Total cholesterol: normal value < 5.16 mmol/L, with a range of 5.16-6.16 mmol/L, and 6.20 mmol/L was considered high. 

(i) LDL cholesterol: normal value < 2.58 mmol/L, with a range 3.35-4.0 mmol/L, and 4.12-4.87 mmol/L was considered high. 

(ii) HDL cholesterol: a value was considered normal if < 1.0 mmol/L and high if ≥ 1.54 mmol/L. 

(iii) Triglycerides: normal value < 1.71 mmol/L and a value of ≥ 2.28 mmol/L was considered high.”

Table I shows the income level of the patients.

Comment 2. The recruitment lasted for 8 months. I would recommend to the authors to recruit more patients given that the Center for Combating Diabetes (CCD) has an average of 12660 patients with diabetes per month. If the circumstances do not allow this add this as a study limitation.

Response 2: This limitation has been addressed in the Limitations of the study section of the manuscript. 

The duration of the data collection period was dictated by several factors: First, the availability of the CCD doctors who work mostly part-time in the center and are hence under significant time constraints. Secondly, the position of the principal investigator (Professor of Neurology at the Faculty of Medicine) allowed us to have the residents conduct their doctoral theses on parts of this study. The doctoral theses also had time constraints. Finally, this work was not funded. Therefore, once further funding is obtained, a large-scale study will allow this obstacle to be overcome.

Comment 3. Correct the power analysis. The previously reported prevalence of DPN by Traore et al. that you used for your sample size calculation was not 50% but 67.7%. So you will have to state that with a population estimate of 12660 in CCD, an estimated prevalence of DPN of 67.7% and margin of error of 6% the minimum sample is 230. It is not recommended to exceed 5% margin of error but with 5% margin of error the minimum sample size is 328, which is higher than your sample size (n=252). I want to emphasize that this power analysis is only to determine the prevalence of DPN.

Response 3: The chosen margin of error was 5% with our 95% confidence interval. Our minimal sample size should be valid. 

Comment 4. It is not enough to assess for association between risk factors like obesity, dyslipidemia with DPN. Mention this as a study limitation.

Response 4: We performed a linear regression analysis between obesity and dyslipidemia with DPN as the dependent variable and we assumed that they were risk factors. This has been added to the Limitations of the study section of the manuscript. 

Comment 5. The paper is poorly written, not only grammatically but also there are incomplete sentences (i.e. aimed to determine the prevalence to describe) and inappropriate use of words or terms (i.e. treatments remain “pregnant” in Africa/ diabetics patients /one for the initial evaluation “on” for present evaluation). Avoid using the term diabetic, instead use patient with diabetes. The writing needs significant improvement.

Response 5: The manuscript has been re-written to address these various issues and errors relating to the grammar and spelling. 

Comment 6. If the combination of obesity and dyslipidemia was associated with DPN, please show in the table in frequency as n (%) of those with obesity and dyslipidemia in the group with/out DPN and provide the p value using chi-square analysis. Avoid repeating the percentage twice in the tables. Why is the percentage of combination of obesity and dyslipidemia 34.7 and 36.6. It should be 36.6%.

Response 6: For Table I, the contingency tables now include the variables requested by the reviewer. We have calculated and presented the OR and the Chi-square statistic with the p-value.

In Table II, the value for the combination of obesity and dyslipidemia is 34.7%.

Comment 7. You show the percentage of different level of education, income, gender in Table 1 but you have not determined what their association with DPN is. Do a chi-square analysis and include the frequency between those with/out DPN and the p value

Response 7: This has now been done, as indicated in Table I.

Comment 8. Was there no association between those been treated with conventional, unconventional (i.e. Voodoo amulet) treatment and no treatment with DPN?

Response 8: No association was found between the use of traditional medical practices and DPN. 

We found an association between the two variables (the type of treatment in general and DPN) using Fisher's exact test, with a p-value = .045. (< 0.05 for statistical significance). 

Comment 9. Were any of the 5 parameters from the quality of life assessment associated with DPN? The reporting of the percentage of people with difficulty with grooming and getting dressed, mobility, anxiety, day-to-day activities are not related to the objective of the study if you have not assessed for their association with DPN. Do a chi-square analysis or remove this data if the numbers between those with/out DPN are not enough. Without this you can’t state in your conclusion that “DPN has a negative impact on the quality of life, independently of the pain”.

Response 9: We have removed this from the conclusion because the chi-square analysis was not applicable (see Table III). Nevertheless, for those who had DPN, there were 108 more patients with autonomy problems than the number of patients without autonomy problems (p = .0001).

Comment 10. I can’t emphasize on how much important is that you keep the analysis focused and simple, remove irrelevant information, shorten the results and discussion and avoid errors. Remove Table 3 because this is not a review paper. Although you do not need to repeat in the results everything written in the Tables you need to be consistent. In the results 38.1% have done primary school but in the table 38.1% are from primary and secondary school. Similar thing with the income level, check the discrepancy about the 66.5% in the results and Table 1.

Response 10: Thank you for highlighting these issues. Table III has been redesigned with the dependent variables DPN+ and DPN-. 

The error in the Results section has been corrected. 

Comment 11. Autonomic neuropathy cannot be defined by one sign only (i.e. urinary retention, erectile disorder or cuts or sores on the feet). It is best to leave this out.

Response 11. This concern about the neuropathy has been addressed in the revised version of the manuscript. 

Comment 12. In the methods you mentioned that HbA1c obtained, HbA1c was not included in the analysis. Why is that? If you have enough HbA1c data you should include it.

Response 12: We decided to not include HbA1c in the analysis because for some of the patients this was performed under questionable technical conditions. We again wish to point out that the biological tests were entirely the responsibility of the patients.

Comment 13. If the Center for Combating Diabetes (CCD) is composed of hospitals or centers inside or outside Bamako mention this in the methods. It is recommended that you include the study design and ethical approval before the assessments.

Response 13: The following paragraph has been inserted at pages 8-9 in the Methods section. 

“…and the Center against Diabetes in Mali (CCD), which comprises the Malian association of diabetic patients (Malian Association for the fight against Diabetes). It is affiliated with the International Diabetes Federation. Its headquarters are in the capital city Bamako, with a focal point in all of the country’s health districts. It represents a basic healthcare facility at the first level in the Malian healthcare pyramid.”

Comment 14. In the authorship, Nadine ATTAL has no affiliation and none of the authors are affiliated with Federal Pain Palliative Care and Support.

Response 14. This has been corrected. We have added the affiliation of Nadine ATTAL and we have now also specified the co-author affiliated with the Federal Pain Palliative Care and Support unit. 

 Reviewer #2: 

The study examined the prevalence of painful and painless diabetic peripheral neuropathy (DPN) in Mali and fills an important gap in literature in this country. 

Thank you for your interest in our work. 

Comment 1. The baseline characteristics data is very poorly presented and a lot of important data is missing -The mean age, weight, BMI, duration of diabetes, HbA1c, BP, and details of lipids panel values are lacking from the baseline characteristics table(1) and Table 11. These are important risk factors that need clear presentation in the main baseline characteristics table. It would be very important to compare between those who were diagnosed with DPN vs those who weren't. The same comparisons need to be made between those who had painful DPN and those who do not have the condition.

Response 1: Table I provides data based on the duration of the diabetes. We have also added the other requested variables. 

Comment 2. The association of risk factors with DPN in Table 11 is not very informative. While obesity was clearly defined with BMI > 30, dyslipidemia has not been defiend in the methods section. 

The way of obtaining the odd ratios was not cleary defined in the statistical methods. Ideally, Binary regression model with DPN as a dichotomous dependent variable and feeding in all risk factors in the model including gender, duration of diabetes, HbA1c, age, plus the other variables included in table 1 and then show the independent predictors of DPN and painful DPN with the odd ratios and P values. 

Response 2: Dyslipidemia has been defined in the Methods section. In this section, we have also added an improved analysis method involving a binary regression model with DPN as a dichotomous dependent variable, as recommended. This type of analysis was carried for the variables in Tables I, II, and III, as well as for the treatment of neuropathic pain (Figure 1) and the conventional treatment of DPN (Figure 2). 

 Comment 3. The authors did not mention any information about the prevalence of foot ulceration and amputation in this sample. This is important to show the magnitude of the problem of the end product of this neglected complication of diabetes.

Response 3: Aspects of the examination of the feet have been specified in Table II. 

Comment 4. While the diagnosis of DPN is a straightforward clinical one and MNSI and DN4 are very robust, its a diagnosis of exclusion . Has B12, folate deficiency and other causes of neuroapathy been excluded ?

Response 4: We wanted to use these robust tools in the absence of other additional examinations. It should be kept in mind that the clinical examination was carried out by neurologists and residents in neurology who were familiar with these tools.

 Comment 5. Its important to mention this in the methodology to acknowledge that in limitations section if not done.

Response 5: Folate and Vitamin B12 deficiencies have not been formally excluded in all of our patients given the financial constraints for some patients. We again wish to point out that, in accordance with the Malian health policy, additional investigations are borne by the patients.

Comment 6. The authors attributed the high prevalence of DPN in this study to the performance of the diagnostic tools used together with the expertise of the team, however, other risk factors such as old age, longer duration of diabetes and poor glycemic control may be contributing factors and need to be mentioned in the discussion.

Response 7: Thank you for this contribution. The sentence “Other risk factors such as the relatively high average age of our patients, a long duration of the disease, and poor glycemic control may also explain the high prevalence of DPN in our study.” has been added to the Discussion section on page 18. 

Comment 7. The only medication mentioned by the authors for painful neuropathy is Amitriptyline, how about the use of other agents mentioned in Table IV? any combinations ? Is it a cost issue alone or individual physician treatment approaches ?

Response 7: As shown in Table IV, the other medications validated in the treatment of neuropathic pain (pregabalin and gabapentin) are not within the reach of our patients due to their non-availability and non-accessibility. Amitriptyline is the only drug validated in this indication that is on the list of essential drugs of the country, which can be partly explained by its low cost and its ready availability in Mali.

Furthermore, we cannot formally exclude problems with training prescribers in the treatment of neuropathic pain. We would then restrict the prescription of other drugs available in private pharmacies in large urban cities in Mali.

Comment 8. The English language needs major editing for grammar and spelling mistakes and sentence formation , ideally by a native English-speaker .

Response 8: This revised manuscript has been edited by a native English speaker. 

The following paragraph has been added on page 24.

“Markers based on electrodiagnosis, sensory system testing, nerve biopsy, skin biopsy, corneal confocal microscopy, pharmacological tests, skin vasomotor reflexes, and pupillometry are not available in Mali. We, therefore, did not use any such markers in this study. . In addition, the data were collected over a relatively short period of time that was independent of us for two main reasons: (i) completion of the study questionnaire took approximately 40 minutes, which was too long for some of the patients and even for a number of physicians from the CCD because they were part-time workers or often volunteers in this center (ii) MD theses had time constraints since the residents in neurology had to be graded by the end of their fourth year. The availability of additional research funds would allow a large-scale study to be undertaken that could potentially overcome these limitations and constraints.”

---

## [Editor Report · Decision Letter 1]

14 Oct 2020

Diabetic polyneuropathy with/out Neuropathic pain in Mali: a cross-sectional study in two reference Diabetes treatment centers in Bamako (Mali), Western Africa

PONE-D-20-18728R1

Dear Dr. MAIGA,

We’re pleased to inform you that your manuscript has been judged scientifically suitable for publication and will be formally accepted for publication once it meets all outstanding technical requirements.

Kind regards,

Rayaz A Malik, MBChB, PhD

Academic Editor

PLOS ONE

Additional Editor Comments (optional):

Most of the major issues have been addressed.
---

## [Editor Report · Acceptance letter]

22 Oct 2020

PONE-D-20-18728R1 

Diabetic polyneuropathy with/out Neuropathic pain in Mali: a cross-sectional study in two reference diabetes treatment centers in Bamako (Mali), Western Africa 

Dear Dr. MAIGA:

I'm pleased to inform you that your manuscript has been deemed suitable for publication in PLOS ONE. Congratulations! Your manuscript is now with our production department. 

Kind regards, 

on behalf of

Professor Rayaz A Malik 

Academic Editor

PLOS ONE